# Projection of the Number of Elderly in Different Health States in Thailand in the Next Ten Years, 2020–2030

**DOI:** 10.3390/ijerph17228703

**Published:** 2020-11-23

**Authors:** Panupong Tantirat, Repeepong Suphanchaimat, Thanit Rattanathumsakul, Thinakorn Noree

**Affiliations:** 1Division of Epidemiology, Disease Control, Ministry of Public Health, Nonthaburi 11000, Thailand; rapeepong@ihpp.thaigov.net (R.S.); nigagape@hotmail.com (T.R.); 2International Health Policy Program (IHPP), Ministry of Public Health, Nonthaburi 11000, Thailand; thinakorn@ihpp.thaigov.net

**Keywords:** bedridden, long-term care, multi-state modelling, elderly, Thailand

## Abstract

The objective of this study is to predict the volume of the elderly in different health status categories in Thailand in the next ten years (2020–2030). Multistate modelling was performed. We defined four states of elderly patients (aged ≥ 60 years) according to four different levels of Activities of Daily Living (ADL): social group; home group; bedridden group; and dead group. The volume of newcomers was projected by trend extrapolation methods with exponential growth. The transition probabilities from one state to another was obtained by literature review and model optimization. The mortality rate was obtained by literature review. Sensitivity analysis was conducted. By 2030, the number of social, home, and bedridden groups was 15,593,054, 321,511, and 152,749, respectively. The model prediction error was 1.75%. Sensitivity analysis with the change of transition probabilities by 20% caused the number of bedridden patients to vary from between 150,249 and 155,596. In conclusion, the number of bedridden elders will reach 153,000 in the next decade (3 times larger than the status quo). Policy makers may consider using this finding as an input for future resource planning and allocation. Further studies should be conducted to identify the parameters that better reflect the transition of people from one health state to another.

## 1. Introduction

The world is now experiencing many aged societies. The definitions of an aged society are multiple and can be categorized into many subgroups, namely, aged, complete-aged, and super-aged societies. An aged society refers to a state where people equal to or more than 60 years constitute over 10% of the total population. Complete-aged and super-aged societies contain a volume of elderly people aged over 60 by more than 20% and 28% of the total population, respectively [1].

Thailand is one among many countries facing aging society challenges. The country has been considered an aged society since 2005, and it may turn to a complete-aged and super-aged society by 2025 and 2030 respectively, if the growth rate of aging population continues at the current pace [2]. A state of dependency of the elderly is one of the critical challenges in the Thai health service system. This challenge is coupled with the high prevalence and incidence of chronic non-communicable diseases (NCDs) such as hypertension, diabetes, stroke, and coronary artery diseases [3,4]. These diseases are likely to undermine the well-being of the elderly and make the elderly prone to a state of to complete dependency, so-called, bedridden.

The Thai Ministry of Public health (MOPH) classified the elderly into three groups, namely, social group, home group, and bedridden group, based on the health status as measured by Activities of Daily Living (ADL). The ADL was an index comprising many health components, such as mobility status, ability to climb up and down the stairs, self-dressing and toileting. The index is scored between 0 and 20. The social group included the elderly with ADL of more than 11, reflecting a group of people who can take care of themselves. The home group included the elderly who can take care of themselves to a certain extent (ADL = 5–11). The bedridden group included the elderly who cannot perform basic living activities by themselves (ADL < 5) [5]. In 2015, MOPH reported that about 79% of the elderly in Thailand (5 million) were in the social group and the rest, 21%, were in home and bedridden groups combined [6]. The National Health Exam Survey also found that 0.2% of the elderly all over the country were in the bedridden group [7]. Projection of health status attracted much attention in many countries, such as studies in Australia (multiple state model) [8], UK (multiple state model) [9], European nations (macro-simulation model) [10], Taiwan (Grey forecasting model) [11], and Sweden (extrapolative model) [12]. In Thailand, there were few studies that attempted to project the volume of elderly in different health status. The latest projection that applied multiple state model in Thailand was in 2014—the situation during that time was far different from the current situation [13]. Thus, updating the current situation of the volume of elderly in different health stages is likely to be helpful for future health policy planning.

One of the main concerns of many health authorities in Thailand, especially the MOPH, is whether and to what extent the Thai healthcare system will have adequate capacity to address the health needs of the elderly in the future. This question cannot be answered unless the projected number of the elderly in the future is first identified. Therefore the objective of this study is to predict the volume of the elderly in different health status categories in Thailand in the next ten years (2020–2030).

## 2. Method

### 2.1. Study Design and Development of Conceptual Framework

This study applies a quantitative secondary data analysis. A multi-state model was applied. Two rounds of consultative stakeholder meetings were conducted on 25 February 2020 and 19 March 2020. The participants of the meeting comprised three policy makers, six academics, and two representatives of the health-systems related institutes. The opinions of the attendees were used as the key inputs to construct the model. An additional literature review was performed to identify key parameters and validate the model assumption [8,9,13,14,15,16].

Participants in the stakeholder meetings helped structure the outline of the model. The model focused on the population aged more than 60 years, and consisted of three parts: incoming population, transition state, and dead state. The incoming population represented the transition of population from people aged younger than 60 years to the cohort of 60 years of age and onwards. The transition state represented the probability of people remaining in the same health status or moving to another health status in each year. For example, the people in the social group could move to home, bedridden, or dead groups, or they could remain in the social group. The dead state was the absorbing state of all populations. The overview of the model framework is presented in Figure 1.

The model equation was based on liner difference equations method. The population in the social group at time t + 1 was calculated from the number in the social group population at time t added to people who transferred from the other groups (home or bedridden) to the social group, subtracting the outgoing population (such as those transferring from the social group to other groups included death). All model equations are presented in Table 1.

### 2.2. Parameter Management

Many parameters were used to estimate the outcome. Λ denoted the volume of population cohort in each year. Δ denoted the proportion of deaths in a specific cohort. Θ denoted the transferal probability between two elder groups, which could be broken down into six scenarios (Θ_SH_ [from social group to home group], Θ_SB_ [from social group to bedridden group], Θ_HS_ [from home group to social group], Θ_HB_ [from home group to bedridden group], Θ_BS_ [from bedridden group to social group], and Θ_BH_ [from bedridden group to home group]). The final outputs were presented in terms of S(t), H(t), B(t), and D(t) which refer to the number of cases in social, home, bedridden, and dead groups at time t, respectively. The time step for the analysis was one year. The analysis commenced when people reached 60 years of age.

#### 2.2.1. Population Cohort in Each Time Frame (Λ)

The volume of population cohort moving from a particular time point to the following time point (that is, number of people started at time (t) and remained in that health status in time (t + 1) was estimated by trend extrapolation methods with exponential growth. Two parameters were used in this step. First, Λ_D_ denoted the mortality rate of new people who died before transferring to the following year; and second, π denoted the prevalence of people in each health status at any time (t − 1). This meant for the first cohort included in the model, π referred to the percentage of people in a particular health status when they were 59 years of age. The mortality rate of people who died before becoming 60 years old was calculated from the following formula: π_die before become 60-year-old_ = (n. of people aged 60 years—n. of people in the same cohort at aged 59 years)/(n. of people in the same cohort at aged 59 years). We obtained the number of people in each specific age group from the Civil Registry website [17]; and we used use the mean mortality rate for each cohort year from the Civil Registry data between 1994 and 2019 [17]. The prevalence of the population in each health status (social, home, and bedridden) was based on the study by Charnduwit in 2017, which calculated the prevalence based on population aged between 60 and 64 years [14]. The equation used to estimate the number of population cohort was shown in Table 2.

#### 2.2.2. Transition Probabilities (Θ)

Transition probability is the likelihood (percentage) that an elder in one particular group (social or home or bedridden group) would change to another in the following year. For example, elderlies in home group in year X faced a chance to change their status to bedridden group in year X + 1; the chance would be labeled as transition probability from home to bedridden group. The transition probabilities across health statuses was acquired by the literature review [9]. We adjusted these parameters against the actual data in 2014 [14,17]. The population in 2014 in social, home, bedridden, and dead groups was 8.8 million, 181,000, 107,000, and 284,000, respectively [14,17]. We used ‘Solve Nonlinear Equations Function’ (generalized reduced gradient [GRG] method) in Microsoft Excel 365 (Microsoft (Thailand) Ltd., Bangkok, Thailand), which intended to identify the most fitted parameters (producing the least errors) in the model. GRG method is one of the widely used optimization methods in many prior studies [18,19].

#### 2.2.3. Starting Population (Pop(t = 0))

The volume of population at the start of the analysis (initial reservoirs) was explored. Pop(0) denoted the population aged ≥ 60 years) in 2019 (time 0). S(0), H(0), and B(0) denoted the volume of people aged ≥ 60 years in the social group, the home group and the bedridden group in 2019, respectively. D(0) denoted the number of deaths in 2019. Π denoted the prevalence in specific groups, which was obtained from the prior study by Charnduwit [14]. We assumed that at the beginning of the analysis, D(0) equated 0. The following formula was used to determine initial reservoirs, Table 3.

#### 2.2.4. Mortality Rate (Δ)

A group-specific mortality rate was determined by the following formula, see Table 4. γ denoted a factor of each health specific group that was used to adjust the crude mortality rate. R denoted relative risk of mortality. The numerators were obtained from the Report of Statistics in the Public Health of Thailand, 2018 [20]. The denominators were retrieved from the Civil Registry. We obtained group-specific relative risk (R) from a Danish nationwide population-based cohort study by Ryg et al. [21]. We used published data (not raw data) of Ryg et al.’s study to calculate the relative risk to serve as an input for our model.

### 2.3. Model Prediction Error

The error of social, home, bedridden, and dead group prediction was calculated by mean absolute percentage error (MAPE). The overall model error was a mean average of the errors in all groups combined (Table 5).

### 2.4. Model Validation

We validated the model by calculating the numbers in social, home, bedridden, and dead groups in 2014 (based on the initial data in 2013) and adjusted this result with the actual data in 2014.

### 2.5. Projection of the Amount of Bedridden Patients

We used the optimized parameters to calculate the numbers of elderly in social, home, bedridden, and dead groups based on the equations shown in Table 5. We set the starting point (t = 1) at 2020 and the end point at 2030.

### 2.6. Sensitivity Analysis

We performed sensitivity analysis by repeating the calculation for 10,000 cycles. The key parameters, such as the prevalence of a specific group for calculated incoming population (π), crude mortality rate (Δ), relative risk of specific group mortality (R), and prevalence of specific group in the elderly (Π), were changed in each cycle, based on their distribution. The transit probability was the main parameter of interest. We presumed four different scenarios with different degree of changes in transit probability (changes by a maximum of 5%, 10%, 15%, and 20%, respectively). Then we assessed the change in the number of bedridden populations by 2030 in these four different scenarios from the base model.

### 2.7. Statistical Software

All analyses were performed by R 4.0 (The R Foundation for Statistical Computing, Vienna, Austria), RStudio^®^ (© 2020 RStudio, Boston, MA, USA), and Microsoft Excel (Microsoft (Thailand) Ltd., Bangkok, Thailand).

## 3. Results

### 3.1. Parameter Identification

Summary of parameters used for projects are presented in Table 6 include population cohort in each time frame, transition probabilities, Starting population, and mortality rate.

#### 3.1.1. Population Cohort in Each Time Frame (Λ)

The population cohort in each time frame was calculated using the exponential equation, presented as Λ = 302,880 × e^0.0359(t)^, where t started from the calendar year of 1994. The coefficient of determination (R^2^) was 0.9442. Parameters that were used to indicate the number of incoming populations were π_S_ = 0.9922, π_H_ = 0.0029 (0.000031), π_B_ = 0.0049 (0.000041) and π_D_ = 0.0131 (0.0054).

#### 3.1.2. Transition Probabilities (Θ)

Transition probabilities of Θ_SH_, Θ_SB_, Θ_HS_, Θ_HB_, Θ_BS_, and Θ_BH_ equated 0.0169, 0.0071, 0.1257, 0.0782, 0.0470, and 0.0688, respectively. The value of these parameters after adjusting with the actual data was 0.0049, 0.0002, 0.1259, 0.0782, 0.0470, and 0.0709, in consecutive order. After adjustment, the model error reduced from 38.67% to 1.40%.

#### 3.1.3. Starting Population

The overall population volume in 2019 was 11,136,059. The prevalence of the elderly in social, home, and bedridden groups was 0.9683, 0.0199 (0.000046), and 0.0118 (0.000036), respectively.

#### 3.1.4. Mortality Rate (Δ)

The relative risk of the mortality rate of the home group in comparison with the social group was 1.45 (95% CI = 1.4529 − 1.4470), while the relative risk of mortality rate of the bedridden group when compared with the social group was 2.27 (95% CI = 2.2749–2.2656). The crude death rate for the overall population was 0.0307 (0.0011).

### 3.2. Model Validation

The prediction error in social, home, and bedridden groups was 0.30%, 0.003%, and 4.95%, respectively. The overall model error, given all group-specific errors combined, was 1.75%.

### 3.3. Projected Amount of Bedridden Patients

The predicted number of the social group population increased from 10.78 million in 2020 to 15.59 million in 2030. The number in the home group also increased (221,610 to 318,980). The number in the bedridden group showed a parabola pattern in the first 4 years followed by a linear increasing pattern from 131,410 to 153,640 thousand between 2020 and 2030. The figures of population estimation in all three groups are presented in Figure 2, Figure 3 and Figure 4.

### 3.4. Sensitivity Analysis

We found that the mean and median of five different scenarios remained unchanged (about 153,000) given different assumptions. The wider the degree of change in transition probabilities, the greater the variation in the number of bed-ridden patients. The volume of bed-ridden patients ranged from 139,110 to 171,616 given the assumption of 20% change in the transition probabilities, while this figure reduced to 142,672 to 161,817 in the base scenario (0% change in the transition probabilities), see Figure 5.

## 4. Discussion

Overall, this study found that the number of bedridden patients would reach 153,640 in 2030; about three-fold larger than the volume of bedridden patients at present. The vast volume of predicted numbers might be due to the large amount of elderly entering the cohort in each year and the large degree of variation of the input parameters.

Our study projected a smaller amount of bedridden patients than the suggested number presented by Chandeovwit et al. which indicated that the number of the patients would be about 240,000 [14]. This difference might be due to the difference in the model assumption and design. The study by Chandeovwit et al. used a constant proportion of dependency rate while our study applied a multistate method. Another study by Srithamrongsawat et al. also projected relatively similar amount of patients to our study; that is, the volume of bed ridden patients in 2024 would reach 110,000 (our study forecasted the figure of 133,000 at the same corresponding time) [13]. The prediction error of this study was small (error = 1.75%). The error appeared to be largest in bedridden group (4.95%).

According to the National Health Statistical Office’s recommendation, a full-time care-manager (CM) should take care of about 35–40 home and bedridden patients and a care-giver (CG) should take care of about 4–8 home and bedridden people [6] Based on these figures, it means that there should be about 10,633–15,039 CM and 53,170–131,597 CG in the health care system to meet the standards set by the NHSO. However, the registered CM and CG in 2020, as reported by Department of Health, numbered only 13,175 and 83,751, respectively [22].

In term of methodology, this study faced some limitations. We used multistate modelling because it better reflects the real situation than the static model [15]. However, the model requires a number of parameters to serve as the model input. The most important input was transit probability. The transit probability in this study derived from a parameter estimation rather than empirical study. Moreover, the empirical study needed to estimate transit probability originating from longitudinal research to better reflect the actual transition of a person from one health state to another [9,15]. However, the longitudinal study of health status of the elderly in Thailand was lacking. Another point of concern is that certain demographic variables, such as economic status and education background, have not been considered in this model [13,14]. Some parameters included in this model were quite outdated but they were the most recent data we could access. In addition, most literature used in this study was from foreign studies which might not reflect the actual healthcare system and population demography in Thailand. This phenomenon reflects gaps in knowledge in the field of mathematical modelling for public health use in Thailand.

Policy makers or relevant academics may use the results in this study as inputs for policy decision-making, particularly in terms of resource planning and mobilization. We recommend that further research should be conducted on specific types of bedridden elderly such as groups of patients with diverse socioeconomic backgrounds. Moreover, there should be additional empirical research on the transition of health status in the population. This will enable researchers and mathematical modelers in the field to access data that better reflect the real situation on the ground, and at the same time benefit the accuracy of model prediction in the future.

## 5. Conclusions

We estimated that, by 2030, the number of bedridden elderly would be about 153,000. This figure is about three times larger than the status quo. Policy makers and academics may consider using this finding as an important input for future resource planning, in light of the coming aged society. Further empirical studies should be conducted to identify the parameters that better reflect the transition of people from one health state to another. Additional research that explores subgroups in populations with different economic backgrounds is recommended.

## Figures and Tables

**Figure 1 ijerph-17-08703-f001:**
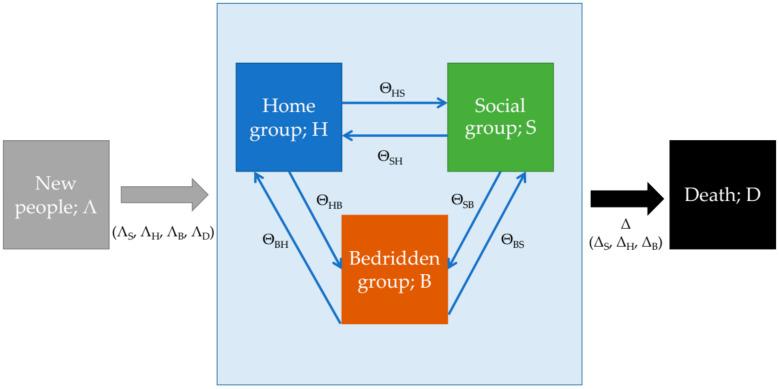
Framework of the model used in this study.

**Figure 2 ijerph-17-08703-f002:**
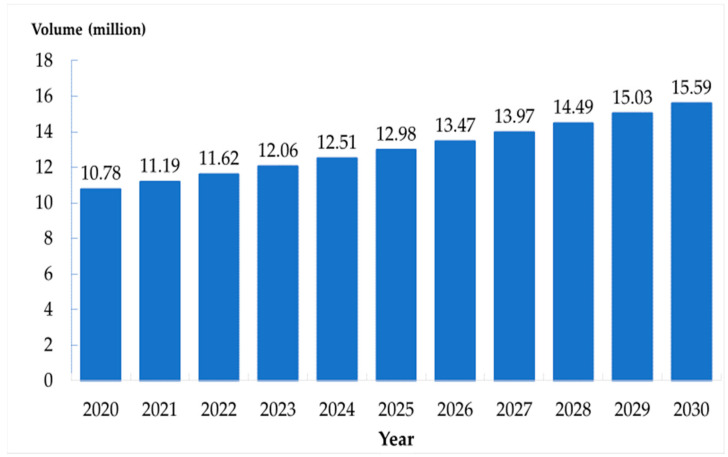
Number of the elderly in social group in Thailand, 2020–2030.

**Figure 3 ijerph-17-08703-f003:**
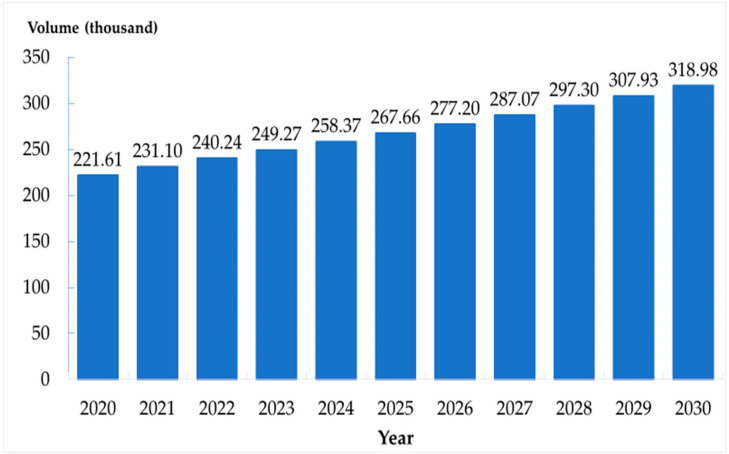
Number of the elderly in home group in Thailand, 2020–2030.

**Figure 4 ijerph-17-08703-f004:**
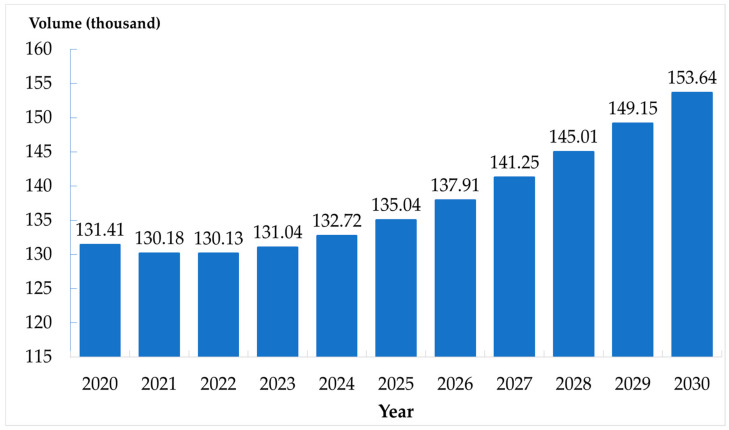
Number of the elderly in bedridden group in Thailand, 2020–2030.

**Figure 5 ijerph-17-08703-f005:**
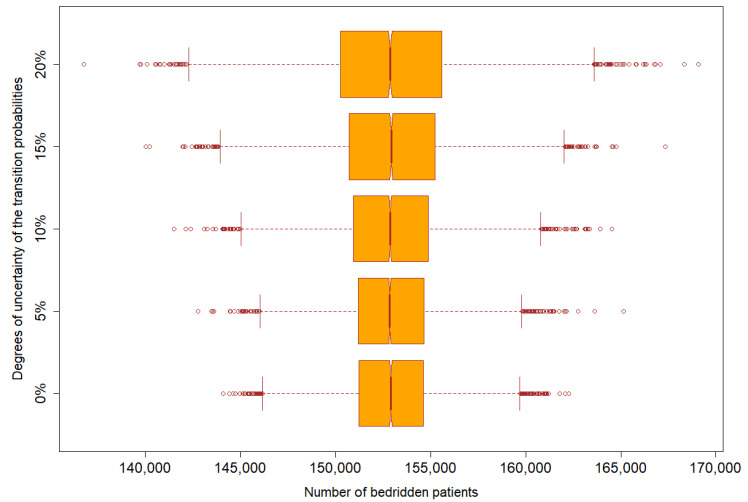
Number of bedridden patients given different degrees of uncertainty of the transition probabilities (scenarios with 5%, 10%, 15%, and 20% changes).

**Table 1 ijerph-17-08703-t001:** Summary of the liner difference equations used to estimate volume of patients in social, home, bedridden, and dead groups.

Model Equation	Formula
1	S(t + 1) = S(t) + Λ_S_(t) + H(t) × Θ_HS_ + B(t) × Θ_BS_ − S(t) × Θ_SH_ − S(t) × Θ_SB_ − S(t) × Δ_S_
2	H(t + 1) = H(t) + Λ_H_(t) + S(t) × Θ_SH_ + B(t) × Θ_BH_ − H(t) × Θ_HS_ − H(t) × Θ_HB_ − H(t) × Δ_H_
3	B(t + 1) = B(t) + Λ_B_(t) + H(t) × Θ_HB_ + S(t) × Θ_SB_ − B(t) × Θ_BS_ − B(t) × Θ_BH_ − B_G_(t) × Δ_B_
4	D(t + 1) = D(t) + S(t) × Δ_S_ + H(t) × Δ_H_ + B(t) × Δ_B_

**Table 2 ijerph-17-08703-t002:** Summary of the equations used to estimate the number of population cohort at time t.

Equation	Formula	Description
1	Λ_D_ = π_die before becoming 60-year-old_ × Λ	Number of deaths at time t − 1
2	Λ′ = Λ − Λ_D_	Number of the elders entering time t
3	Λ_S_ = π_S_ × Λ′	Number of social group entering time t
4	Λ_H_ = π_H_ × Λ′	Number of home group entering time t
5	Λ_B_ = π_B_ × Λ′	Number of bedridden group entering time t

**Table 3 ijerph-17-08703-t003:** Summary of the equations used to estimate the volume of population at the start of the analysis (initial reservoirs).

Equation	Formula	Description
1	Pop(0) = S(0) + H(0) + B(0) + D(0)	Total volume of population aged ≥60 years consisted of people aged ≥60 years in the social group, the home group, the bedridden group, and the death group at time 0.
2	S(0) = Pop(0) × Π_S_	Volume of people aged ≥60 years in the social group resulted from population aged ≥60 years multiplied by prevalence of social group.
3	H(0) = Pop(0) × Π_H_	Volume of people aged ≥60 years in the home group resulted from population aged ≥60 years multiplied by prevalence of home group.
4	B(0) = Pop(0) × Π_B_	Volume of people aged ≥60 years in the bedridden group resulted from population aged ≥60 years multiplied by prevalence of bedridden group.
5	D(0) = 0	Volume of dead people in the model. We assumed there was no death at the beginning of the analysis.

**Table 4 ijerph-17-08703-t004:** Summary of the equations used to estimate group-specific mortality rate.

Equation	Formula	Description
1	$\Delta =\frac{\Pi_{s}\Delta_{S}{+\;\Pi }_{H}\Delta_{H}{+\;\Pi }_{B}\Delta_{B}}{\Pi_{S}+\Pi_{H}+\Pi_{B}}$	Crude mortality rate was a prevalence weight average of group-specific mortality.
2	Δ_S_ = $\gamma_{s}$ × Δ	Social group mortality rate was social group specific severity factors multiply by crude mortality.
3	Δ_H_ = $\gamma_{H}$ × Δ	Home group mortality rate was home group specific severity factors multiply by crude mortality.
4	Δ_B_ = $\gamma_{B}$ × Δ	Bedridden group mortality rate was bedridden group specific severity factors multiply by crude mortality.
5	$R_{H}=\frac{\Delta_{H}}{\Delta_{S}}=\frac{\Delta \times \gamma_{H}}{\Delta \times \gamma_{s}}$	Relative risk of home mortality was calculated from home group mortality rate over social group mortality rate.
6	$R_{B}=\frac{\Delta_{B}}{\Delta_{S}}=\frac{\Delta \times \gamma_{B}}{\Delta \times \gamma_{s}}$	Relative risk of bedridden mortality was calculated from bedridden group mortality rate over social group mortality rate.
7	$\gamma_{s}=\frac{1}{\Pi_{S}{+\Pi }_{H}R_{H}{+\Pi }_{B}R_{B}}$	Social group mortality was calculated from group specific prevalence and relative risk. This equation was rewritten form of equations 1–6.

**Table 5 ijerph-17-08703-t005:** Summary of the equations used to calculate model error.

Equation	Formula	Description
1	$Error_{j}=\frac{1}{N}\overset{N}{\underset{i=1}{\sum }}\frac{\left|Observerd_{ij}-Predicted_{ij}\right|}{Observerd_{ij}}$	Mean absolute percentage error (MAPE) of group j was the summation of absolute difference divided by observed value.
2	$Model\;error=\frac{1}{K}\overset{K}{\underset{j=1}{\sum }}Error_{j}$	Model error was mean average of all group-specific errors combined.

**Table 6 ijerph-17-08703-t006:** Summary of parameters used for projecting the amount of bedridden patients.

No	Group of Variables	Parameters	Mean	SD	Reference (Ref)
1	New population	Λ	Λ = 302,880 × e^0.0359(t)^		Bureau of Registration administration [17]
2	Prevalence of specific group who age equal 59-year-old	π_S_	0.9922	-	Model calibration from Bureau of Registration administration [17]
3		π_H_	0.0029	0.000031	Charnduwit [14]
4		π_B_	0.0049	0.000041	
5		π_D_	0.0131	0.0054	Model calibration from Bureau of Registration administration [17]
6	Mortality rate	Δ	0.0307	0.0011	Bureau of Registration administration and Strategy and Planning Division [17,20]
7	Relative mortality rate	R_H_	1.45	0.0010 (SE of ln RR)	Ryg [21]
8		R_B_	2.27	0.0010 (SE of ln RR)	
9	Prevalence of specific group in elderly	Π_S_	0.9683	-	Model calibration from Charnduwit [14]
10		Π_H_	0.0199	0.000046	Charnduwit [14]
11		Π_B_	0.0118	0.000036	
12	Mortality rate in specific group	Δ_S_	0.0494	-	Model calibration from Bureau of Registration administration, Strategy and Planning Division, and Ryg [17,20,21]
13		Δ_H_	0.1465	-	
14		Δ_B_	0.2050	-	
15	Transit probability from social group	Θ_SH_	0.0169	-	Model calibration from Rickayzen [9]
16		Θ_SB_	0.0071	-	
17	Transit probability from home group	Θ_HS_	0.1257	-	
18		Θ_HB_	0.0782	-	
19	Transit probability from bedridden group	Θ_BS_	0.0470	-	
20		Θ_BH_	0.0688	-	
21	Initial Total population	Pop(t = 0)	11,136,059	-	Bureau of Registration administration [17]

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
