# Peer review of "Projection of the Number of Elderly in Different Health States in Thailand in the Next Ten Years, 2020–2030"

_ijerph, 2020, doi:10.3390/ijerph17228703_

Round 1

Reviewer 1 Report

Demographic aging has been one of the most discussed aspects of population development in recent decades. It has been rapid in Thailand. Thus, the paper undertakes an important problem. However,  the article needs major corrections.

Detailed remarks for Authors.

  1. There is no literature review of what projection methods of the number of elderly are used in other countries.
  2. The data and the methods are not presented and analyzed in a systematic manner. In particular, in the results section (in line 156) Authors write that “The model equation was based on a differential equation method”. This methodological aspects should be clearly explained in “Method” section. Moreover, the data should be clearer described. For example, the information about data used in exponential trend is the results section (in line 166). It should be done in “Method” section.
  3. The paper is written somewhat carelessly. In particular:
  • in Table 2, the description of the formula for the first equation omits D(0);
  • in “Reference” section instead of “Rickayzen BD, Walsh DEP. A Multi-State Model of Disability for the United Kingdom: Implications for Future Need for Long-Term Care for the Elderly. Vol 8.; 2002. doi:10.1017/s1357321700003755” should be “Rickayzen, B.; Walsh, D. A Multi-State Model of Disability for the United Kingdom: Implications for Future Need for Long-Term Care for the Elderly. British Actuarial Journal, 2002, 8(2), 341-393.”
  • Woranan et al. and Samrit et al. mentioned in the discussion section are omitted in the “Reference” section.

Author Response

Response to reviewer

Reviewer 1

  1. Demographic aging has been one of the most discussed aspects of population development in recent decades. It has been rapid in Thailand. Thus, the paper undertakes an important problem. However, the article needs major corrections.

Thank you for your comments that help improve our study.

  1. There is no literature review of what projection methods of the number of elderly are used in other countries.

We have added some text about prior literature in other countries that aim to project number of the elderly with by different methods. Please see line 54.

  1. The data and the methods are not presented and analyzed in a systematic manner. In particular, in the results section (in line 156) Authors write that “The model equation was based on a differential equation method”. This methodological aspects should be clearly explained in “Method” section.

We rearranged part of the results (the model concept) into the method part. And we changed the description of model equation from ‘differential equation method’ to ‘liner difference equations method’ as per your notice. Please see line 76 for rearranged part and line 86 for liner difference equations method.

  1. Moreover, the data should be clearer described. For example, the information about data used in exponential trend is the results section (in line 166). It should be done in “Method” section.

Thank you for your suggestion. However, we consider showing model parameters in the results is more appropriate since part of our methods is to identify ‘value’ of parameters from model fitting.

  1. The paper is written somewhat carelessly. In particular: in Table 2, the description of the formula for the first equation omits D(0);

Thank you for your comment. We added more detail about D(0) and add more description of equation 1. The whole text throughout the whole document was rechecked again. Please see line 141 Equation 1 and 5.

  1. in “Reference” section instead of “Rickayzen BD, Walsh DEP. A Multi-State Model of Disability for the United Kingdom: Implications for Future Need for Long-Term Care for the Elderly. Vol 8.; 2002. doi:10.1017/s1357321700003755” should be “Rickayzen, B.; Walsh, D. A Multi-State Model of Disability for the United Kingdom: Implications for Future Need for Long-Term Care for the Elderly. British Actuarial Journal, 2002, 8(2), 341-393.”

Thank you for your recommendation. Having checked the citation again in Mendeley software, we found that the mentioned reference should be written as “Rickayzen, B. D.; Walsh, D. E. P. A Multi-State Model of Disability for the United Kingdom: Implications for Future Need for Long-Term Care for the Elderly. Br. Actuar. J. 2002, 8 (2), 341–393. https://doi.org/10.1017/S1357321700003755.” However, we will check all of the reference formats again with the editors.

  1. Woranan et al. and Samrit et al. mentioned in the discussion section are omitted in the “Reference” section.

Sorry for this error. Our error is we wrote down the author’s first name instead of his/her surname. Now, this error is corrected. Please see REF 13 and 14.

Reviewer 2 Report

  1. I could not understand “transition probability”. Some explanation could be added. Different terms (transferring probability, transitional probability, and transition probability). They could be consistent.
  2. Is it possible to add the terms of parameters in Figure 1. And I don’t prefer the three-dimension figures.
  3. Figure 2, 3, and 4 could be combined. The bar chart might be better.

Author Response

Response to reviewer

Reviewer 2

  1. I could not understand “transition probability”. Some explanation could be added. Different terms (transferring probability, transitional probability, and transition probability). They could be consistent.

Thank you for your recommendation. We used the term “transition probability” as it had ever been cited in prior study. We add more explanation of transition probability in line 122. This term is now consistently used throughout the paper.

  1. Is it possible to add the terms of parameters in Figure 1. And I don’t prefer the three-dimension figures.

Thank you for your suggestion. The figure was changed as your wish. Please see line 84.

  1. Figure 2, 3, and 4 could be combined. The bar chart might be better.

Thank you. The line charts were changed to bar chart. However, we preferred separating these into 3 figures because the unit of each figure varied considerably.

Reviewer 3 Report

The objective of this study was to project the number of bedridden elders in Thailand in the 15 next ten years. The results reported by the study are extremely important and necessary for actions and policies to be implemented in time to minimize impacts. I congratulate the authors for the study.

I recommend checking the citation in line 216.

Author Response

Response to reviewer

Reviewer 3

  1. The objective of this study was to project the number of bedridden elders in Thailand in the 15 next ten years. The results reported by the study are extremely important and necessary for actions and policies to be implemented in time to minimize impacts. I congratulate the authors for the study.

We are hugely grateful for your comments

  1. I recommend checking the citation in line 216.

Thank for spotting this. The error in the citation is now corrected; please see REF 14.

Reviewer 4 Report

The Abstract must be modified, at least the first sentence of the same (rows 14-15), since the text of the same is not completely corresponding with the title of the work. I understand that the last sentence of the Introduction (rows 58-60) is more appropriate instead.

In 1. Introduction, the citation to reference 5 appears twice, in row 48 and in row 51. The last one is enough.

Regarding 2. Method, there are five things that concern me and that authors should review

  • It should be noted that the nomenclature used is not at all standard, at least in the literature on health, demography or biometric-actuarial studies.
  • the use of 'Solve Nonlinear Equation Function' Microsoft Excel 365, whose reliability often raises doubts (The results should be contrasted with other mathematical optimization software). 
  • the date the model is validated, 2014, seems too far from the projection start date, 2020.
  • when in row 116 they refer to "a Danish nationwide population-based cohort study", I assume it is to the formulas in group-specific relative-risk (R) and not the data. They should make it clear.

In 3. Results, there are a couple of details that reveal that the authors are not experts in stochastic processes.

  • In rows 151-152, it is technically normal to add that the dead state is an absorving-state, beyond indicating that it is "the final state of all populations."
  • In row 156, the model equation was base on "a finite difference equation method", not in "a differential equation method". It is a discrete model and not a continuous model.

In 4. Discussion, there are two citations that are not in references or do not correspond to them: "Worawan et al." and "Samrit el al.". In this case, if the citations are to references 11 and 8, they should be “Charnduwit” and Srithamrongsawat el al. ”. Otherwise, they are wrong.

On the other hand, socio-economic status greatly influences the health status of the elderly, I encourage them to continue along this line.

In References, the authors should make an effort to update access to data sources. Some are even excessively lagged, dating back to 2003 (rows 54 and 283). Some of the references are incomplete (8,9,11 and 15), essential information is missing.

Author Response

Response to reviewer

Reviewer 4

  1. The Abstract must be modified, at least the first sentence of the same (rows 14-15), since the text of the same is not completely corresponding with the title of the work. I understand that the last sentence of the Introduction (rows 58-60) is more appropriate instead.

Thank you for your recommendation. The abstract was revised. Please see line 13.

  1. In Introduction, the citation to reference 5 appears twice, in row 48 and in row 51. The last one is enough.

Thank you for your recommendation. The duplicated citation was removed.

Regarding 2. Method, there are five things that concern me and that authors should review

  1. It should be noted that the nomenclature used is not at all standard, at least in the literature on health, demography or biometric-actuarial studies.

Thank you for your suggestion. We found that some terms are normally used in many studies (such as the term ‘bedridden’). However we admit that some terms (such as home and social groups) were not a standard text; but we intend to keep these terms as they were already mentioned in the stakeholder meeting (section 2.1). Thus keeping this term has significant benefit in communicating with policy makers.

  1. The use of 'Solve Nonlinear Equation Function' Microsoft Excel 365, whose reliability often raises doubts (The results should be contrasted with other mathematical optimization software). 

Thank you. We found that ‘'Solve Nonlinear Equation Function by Microsoft Excel 365’ was actually ‘generalized reduced gradient (GRG)]—a method which was well supported by prior literature. We have added more details of ‘GRG’ in line 128.

  1. The date the model is validated, 2014, seems too far from the projection start date, 2020.

The data in 2014 was the most updated data that we could access. We admit that it is quite lagged behind, but in another angle, it reflects that research in this issue is in dire need.

  1. when in row 116 they refer to "a Danish nationwide population-based cohort study", I assume it is to the formulas in group-specific relative-risk (R) and not the data. They should make it clear.

Thank you for your recommendation. we add some text to make it clearer, please see line 148.

In 3. Results, there are a couple of details that reveal that the authors are not experts in stochastic processes.

  1. In rows 151-152, it is technically normal to add that the dead state is an absorbing-state, beyond indicating that it is "the final state of all populations."

We agree with your point. And we now changed this term as per your recommendation. Please see line 82.

  1. In row 156, the model equation was based on "a finite difference equation method", not in "a differential equation method". It is a discrete model and not a continuous model.

We changed the description of model to ‘liner difference equations’ to make it correctly specify that discrete model was used instead of a continuous model. Please see line 86.

  1. In  Discussion, there are two citations that are not in references or do not correspond to them: "Worawan et al." and "Samrit el al.". In this case, if the citations are to references 11 and 8, they should be “Charnduwit” and Srithamrongsawat el al. ”. Otherwise, they are wrong.

Sorry for this error. Our error is we wrote down the author’s first name instead of his/her surname. Now this error is corrected. Please see REF 13 and 14.

  1. On the other hand, socio-economic status greatly influences the health status of the elderly, I encourage them to continue along this line.

We agree with your recommendation. However, we lacked detailed data of socioeconomic status of the elderly to perform stratified projection. We have added this point in the discussion paper, pointing that future study in this issue is in dire need. Please see line 257.

  1. In References, the authors should make an effort to update access to data sources. Some are even excessively lagged, dating back to 2003 (rows 54 and 283). Some of the references are incomplete (8,9,11 and 15), essential information is missing.

Thank you for your recommendations. The data in 2003 was a bit lagged behind. However, data on the national exam survey in 2003 was the most recent dataset that contains adequate information to project the volume of bedridden patients. The following surveys reported percentage of people with different Instrumental Activities of Daily Living (IADL) levels, but not the percentage of bedridden patients. Other incomplete references were corrected, for example REF 8, 13, 14, and 20.

Round 2

Reviewer 1 Report

The revised version of the article adequately addresses most of my concerns: the Authors managed to improve it significantly.